# Multi-Trait Single-Step Genomic Prediction for Milk Yield and Milk Components for Polish Holstein Population

**DOI:** 10.3390/ani13193070

**Published:** 2023-09-29

**Authors:** Hasan Önder, Beata Sitskowska, Burcu Kurnaz, Dariusz Piwczyński, Magdalena Kolenda, Uğur Şen, Cem Tırınk, Demet Çanga Boğa

**Affiliations:** 1Department of Animal Science, Ondokuz Mayis University, Samsun 55139, Türkiye; 20280583@stu.omu.edu.tr; 2Department of Animal Biotechnology and Genetic, Faculty of Animal Breeding and Biology, Bydgoszcz University of Science and Technology, 85084 Bydgoszcz, Poland; beatas@pbs.edu.pl (B.S.); darekp@pbs.edu.pl (D.P.); kolenda@pbs.edu.pl (M.K.); 3Department of Agricultural Biotechnology, Ondokuz Mayis University, Samsun 55139, Türkiye; ugur.sen@omu.edu.tr; 4Department of Animal Science, Iğdır University, Iğdır 76000, Türkiye; cem.tirink@igdir.edu.tr; 5Department of Chemistry and Chemical Processing, Osmaniye Korkut Ata University, Osmaniye 80050, Türkiye; demetcanga@osmaniye.edu.tr

**Keywords:** multi-trait prediction, single-trait prediction, milk composition, GBLUP, genomic selection

## Abstract

**Simple Summary:**

The objective of our study was to evaluate the predictive ability of a multi-trait genomic prediction model to estimate heritability and genetic correlations of traits such as 305-day milk yield, milk fat percentage, milk protein percentage, milk lactose percentage, and milk dry matter percentage in the Polish Holstein population. Results showed that strong accuracies for the predictions were achieved. The genetic relations with milk yield were negative, as expected, because increasing milk yield decreases the milk components percentage. In conclusion, multi-trait genomic prediction can be more beneficial than single-trait genomic prediction.

**Abstract:**

The objective of our study was to evaluate the predictive ability of a multi-trait genomic prediction model that accounts for interactions between marker effects to estimate heritability and genetic correlations of traits including 305-day milk yield, milk fat percentage, milk protein percentage, milk lactose percentage, and milk dry matter percentage in the Polish Holstein Friesian cow population. For this aim, 14,742 SNP genotype records for 586 Polish Holstein Friesian dairy cows from Poland were used. Single-Trait-ssGBLUP (ST) and Multi-Trait-ssGBLUP (MT) methods were used for estimation. We examined 305-day milk yield (MY, kg), milk fat percentage (MF, %), milk protein percentage (MP, %), milk lactose percentage (ML, %), and milk dry matter percentage (MDM, %). The results showed that the highest marker effect rank correlation was found between milk fat percentage and milk dry matter. The weakest marker effect rank correlation was found between ML and all other traits. Obtained accuracies of this study were between 0.770 and 0.882, and 0.773 and 0.876 for MT and ST, respectively, which were acceptable values. All estimated bias values were positive, which is proof of underestimation. The highest heritability value was obtained for MP (0.3029) and the lowest heritability value was calculated for ML (0.2171). Estimated heritability values were low for milk yield and milk composition as expected. The strongest genetic correlation was estimated between MDM and MF (0.4990) and the weakest genetic correlation was estimated between MY and ML (0.001). The genetic relations with milk yield were negative and can be ignored as they were not significant. In conclusion, multi-trait genomic prediction can be more beneficial than single-trait genomic prediction.

## 1. Introduction

In general, animal breeding aims to genetically improve the next generation in order to increase the quantity and quality of economic traits in certain environments [1]. Traditional genetic improvement in farm animals has successfully predicted breeding values using phenotypes and pedigree information. Unlike traditional animal breeding methods, genomic selection can estimate breeding values more accurately [2]. Hence, the decision to select animals for breeding could be made based on the joint merit of all SNP markers across the genome [3,4,5].

Genomic selection (GS) has made a significant impact on animal breeding. Unlike traditional selection methods, GS estimates the genetic value of individuals with all genotypes, even when the phenotype has not been determined, based on estimates of marker effects [6]. Instead of using a limited number of markers as in GS, genomic estimated breeding value (GEBV) uses the genomic prediction (GP) of additive genetic traits of animals considered for selection [7,8]. Different statistical models for GP were first described by Meuwissen et al. [4], who evaluated the performance of linear mixed models and Bayesian mixture models in evaluating marker effects and GEBVs [8]. With the introduction of high-density (including >50,000 markers) SNP chips in 2008, genomic selection began. This technology was successfully implemented in many countries. Adapting breeding programs based on genomic selection in major milk-producing countries has led to significant worldwide changes in the dairy industry [4].

Genomic selection is a breeding technique used in the dairy cattle industry (as in other species for the same or different aims) to improve the genetic potential of cattle for milk production and other desirable traits. This technique involves analyzing the genetic information of individual animals using high-throughput sequencing technologies to identify variations in their DNA sequence that are associated with specific traits of interest [9,10].

The process of genomic selection involves the steps of DNA samples collection: sequencing of DNA, genotyping, phenotyping, prediction of breeding values, and selection of breeding candidates [4,7,9,10,11].

Overall, genomic selection allows for a more accurate and efficient selection of breeding candidates as it allows for the identification of genetic markers that are associated with traits of interest even before the animals have been evaluated for those traits. This can lead to faster genetic progress and improved productivity in the dairy cattle industry. Using the genomic selection, the estimation accuracy of breeding values is increased by about 30%, the generation interval is reduced effectively, the possibility for the estimation of low heritability traits increases, the number of candidate sires for progeny testing decreases, and more robust mating programs [7,8,9,10,11] are facilitated.

A great majority of the genomic selection studies [12,13,14,15] have focused on single-trait breeding value estimation. However, multi-trait methods are expected to yield more accurate predictions than single-trait methods as in traditional BLUP. Simulations have shown that multi-trait genomic prediction can lead to a considerable increase in genomic prediction accuracy [16,17,18,19]. Jia and Jannink [16] compared multi-trait (MT) and single-trait (ST) estimation accuracies and mentioned that MT and ST methods showed similar prediction accuracies; however, Guo et al. [17] reported that the multi-trait predictions can improve the accuracy of single-trait predictions for genomic prediction.

There are many methods used for genomic prediction. The Genomic Best Linear Unbiased Prediction (GBLUP) method fitted all of the SNPs into the model, assuming that every SNP explained an equal proportion of the total genetic variance. This method is called Bayes C0 for executive simplicity. It is the same as Bayes C when pi = 0 [7]. The GBLUP method is generally used for genomic prediction in animal breeding, mainly due to its straightforward implementation using the existing computer software. GBLUP also demands a shorter computation time [7,18,20].

The objective of our study was to evaluate the predictive ability of a multi-trait genomic prediction model that accounts for interactions between marker effects to estimate heritability and genetic correlations of traits such as 305-day milk yield, milk fat percentage, milk protein percentage, milk lactose percentage, and milk dry matter percentage in the Polish Holstein population. The hypothesis of this study was that a multi-trait genomic prediction model is more accurate than a single-trait genomic prediction model.

## 2. Materials and Methods

### 2.1. Materials

The genetic material for the study was collected during a routine estimation of breeding value. Genotyping using customized EuroGenomics microarrays was performed by the Polish Federation of Cattle Breeders and Dairy Farmers (PFCB&DF, Warsaw, Poland) using: Eurogenomics v3_POL; Eurogenomics v4_POL; Eurogenomics v5_POL; Eurogenomics v6_POL; Eurogenomics v8b_POL; and Eurogenomics MD_POL with Illumina 15k SNP chips protocol. Because many chips were employed, the same genes from all chips were selected to work. Traits such as 305-day milk yield (MY, kg), milk fat percentage (MF, %), milk protein percentage (MP, %), milk lactose percentage (ML, %), and milk dry matter percentage (MDM, %) were examined, and 14,742 SNP genotype records (after selection on same genes) for 586 Polish Holstein Friesian (PHF) dairy cows reared in 12 different private enterprises in Poland were used.

Milk recording data were collected in the period from 2016 to 2019 according to the A4 method (the interval between two successive recordings of two daily milkings ranging between 28 and 33 days) The data regarding milk performance was collected during routine test days conducted by the Polish Federation of Cattle Breeders and Dairy Farmers (PFCB&DF). Authors were granted access to the data. The description provided in the manuscript is in accordance with PFCB&DF descriptions [21].

### 2.2. Methods

#### 2.2.1. Quality Control and Data Preparation

The quality control of genotypes assumed that the minor allele frequency was greater than 5% (this eliminated monomorphic genes). Animals with SNP missing rates greater than 10% were also excluded from the analysis. After the quality control tests, 497 animals and 12,906 SNPs were used for the genomic prediction [12,13].

#### 2.2.2. Data Analysis

In the predictions, BMTME and BGLR packages were used for the analysis with 20,000 burn-in and 100,000 test iterations. The optimization was achieved successfully. The deviation of the first 20,000 cycles was discarded as burn-in and excluded from the experiment [7]. A total of 400 animals were used as a training population set and the remaining 97 animals were used as a test population set. For the genomic prediction, the population was divided into two groups as train and test populations. The records of the training set were used to build up the estimation model and the records of the test set (not included in the estimation) were used to predict their GEBV and to estimate accuracies [7].

The lactation order (OL, 1, 2 and 3) and farms (1,2,…,12) were used as fixed factors and days in milk (DIM) as a random factor. To examine the relationships among marker effects impact level of traits, Spearman rank correlation analysis was used. To examine the relationships among phenotypic correlations of the traits Pearson correlation analysis was used. Genetic correlation between traits *t*1 and *t*2 was calculated as $\sigma_{g_{t1t2}}/\sqrt{\sigma_{g_{t1t1}}\sigma_{g_{t2t2}}}$ where $\sigma_{g}$ is the genetic variance–covariance matrix for multiple traits. The $\sigma_{g}$ was calculated as $\left(\sum_{k=k1}^{k2}\sum_{i=1}^{p}var({SNP}_{i})a_{i}a_{i}^{T}\right)/\left(k_{2}-k_{1}+1\right)$, where *var*(*SNPi*) is the genotype variance for *SNPi* and $a_{i}$ is the estimated marker effect vector for *SNPi* in iteration *k* for an analysis run over *k*2 iterations and with *k*1 burn-in iterations [16]. Standard error of the heritability was calculated from the iterations [7]. The prediction accuracy was defined as the Pearson correlation between the true breeding values (observed phenotype data) and the predicted GEBV values in the validation population [7].

#### 2.2.3. Single-Trait and Multi-Trait Genomic Prediction Models

The single-trait GBLUP can be defined as given in Equation (1):(1)y=Xb+Za+e
where ***y*** is a phenotype vector for the traits; ***X*** and ***Z*** are the design matrices; ***b*** and ***a*** denote the fixed effects and the additive genetic effects, respectively, and ***e*** is the random error. It is assumed that *a*~*N*(0, *Gσa^2^*) where *σa^2^* is additive genetic variance, *G* is the genomic relationship matrix which consists of SNP marker information. A detailed description of how *G* is computed can be found in Karaman et al. [18].

The multi-trait model (Equation (2)) can be defined as:(2)y=μ+Xb+Za+e
where ***y*** is a vector of length *n* × *t*, (*n* is the number of individuals and *t* is the number of traits), ***μ*** is the mean vector of length *n* × *t*, ***X*** is a block diagonal matrix with fixed effect design matrices per trait on the diagonal and for each trait the design matrix is the same, ***b*** is the matrix of fixed effects of *f* × *t* dimension (*f* is number of fixed effects and *t* is number of traits), ***Z*** is a block diagonal matrix with random effect design matrices per trait on the diagonal. The random effect design matrices are equal for each trait. ***a*** is a vector of predicted genetic values of the individuals for all traits with *a~N*(0, *Σ⊗K*), and ***e*** is the error vector with ***e****~N*(0, *R⊗I)*, where ***K*** is the realized additive relationship matrix among individuals estimated from the markers, ***Σ*** and ***R*** are the unstructured variance–covariance matrices for the genetic and residual effects between traits, respectively [22,23].

## 3. Results

### 3.1. Marker Effects

Marker effects for 305-days milk yield (MY, kg), milk fat percentage (MF, %), milk protein percentage (MP, %), milk lactose percentage (ML, %), and milk dry matter percentage (MDM, %) are given in Figure 1, Figure 2, Figure 3, Figure 4 and Figure 5.

As expected, many of the markers had minor effects on the five traits of interest namely 305-day milk yield, milk fat percentage, milk protein percentage, milk lactose percentage, and milk dry matter percentage (Figure 1, Figure 2, Figure 3, Figure 4 and Figure 5). To examine the relationships among the marker effects of traits, Spearman rank correlations were obtained (Table 1).

The highest marker effect rank correlation was found between MF and MDM. The weakest marker effect rank correlation was found between ML and all other traits. As expected, MF, MP and MDM had negative relations with MY even though the correlation coefficients were very low. These Spearman rank correlation coefficients indicated that the effecting markers of each trait are different. In this situation, multi-trait breeding aims are hard to reach.

### 3.2. Accuracy

The accuracy (r) and deviations (b) of the traits from multi-trait (MT) and single-trait (ST) predictions were calculated and are presented in Table 2.

The highest accuracy was observed for MDM from multi-trait and single-trait prediction. The lowest accuracy was observed for ML from multi-trait and single-trait prediction. The lowest bias was observed for ML. The accuracies calculated from MT prediction had higher accuracies than ST except for ML. The differences in the accuracies between MT and ST were not extremely high.

### 3.3. Genetic Parameters

The highest heritability value was obtained for MP and the lowest heritability value was calculated for ML (Table 3). When the strongest genetic correlation was calculated between MF and MDM (0.499), the weakest genetic correlation was calculated between MY and ML (0.001). As expected, genetic correlations with MY were generally found to be negative except for ML which was nearly zero (Table 3).

## 4. Discussion

Accuracies obtained in our study were between 0.77 and 0.882, and 0.773 and 0.876 for MT and ST, respectively. Guo et al. [17] reported that the accuracies were 0.952 and 0.951 for MT and ST, respectively, with simulated data in which the heritability value was 0.30 and genetic correlation was 0.50. As they mentioned, and is supported by our results, the multi-trait predictions can improve the accuracies of single-trait predictions for genomic prediction. Their high accuracy values over 0.90 may be the result of the simulated data. Mehrban et al. [24] compared ST-ssGBLUP and MT-ssGBLUP using carcass traits (backfat thickness, carcass weight, eye muscle area, marbling score, and yearling weight) of Hanwoo cattle. They found the accuracies 0.45 and 0.48 for ST-ssGBLUP and MT-ssGBLUP, respectively, and the heritability values ranged from 0.48 to 0.59. Our accuracy results were higher even than our heritability values, although lower than Mehrban et al.’s [24] declared results, but supported our results suggesting that MT accuracy was higher than ST accuracy. However, in Mehrban et al.’s [24] results, the MT-ST accuracy differences were higher than in our study. Sandhu et al. [25] declared that the accuracy of MT prediction is 7.9% higher than ST prediction for quality traits in winter wheat. The average accuracy of ST prediction was 0.615 and MT was 0.650. This supports our results in terms of the superiority of MT to ST, and in our study multi-trait prediction showed 0.5% better accuracy compared to the single-trait prediction. Bhatta et al. [26] compared MT and ST prediction using agronomic and malting quality traits in barley. They mentioned that the accuracy of MT prediction is 61% higher than ST prediction for malting quality traits. MT is superior to ST according to our results and the other literature [24,25]. This difference may be due to multi-environmental situations and the heritability differences. Guo et al. [27] compared Multi-environment Genomic Best Linear Unbiased Predictor (MGBLUP), Bayesian Multi-trait Multi-environment (BMTME), Bayesian Multi-output Regressor Stacking (BMORS), Single-trait Multi-environment Deep Learning (SMDL), and Multi-trait Multi-environment Deep Learning (MMDL). They mentioned that the multi-trait model showed 5 to 22% more accuracy compared to the single-trait model for the yield-related traits in soft wheat. Cheng et al. [28] compared MT and ST prediction accuracies on five simulated traits and argued that MT and ST methods showed similar prediction accuracies; their results were very similar to our results. Jia and Jannink [16] used simulated data to compare MT and ST estimation accuracies. They found that MT and ST methods showed similar prediction accuracies, while MT accuracies were slightly higher than ST accuracies. Makgahlela et al. [29] compared MT and ST prediction methods using Nordic Red dairy cattle (Swedish red, Finnish Ayrshire, Norwegian red, and combined breeds) to examine milk, protein, and fat yield. They reported that the superiority of MT prediction to ST prediction on the accuracies were 1.6%, 3.2% and 0% for milk, protein and fat, respectively, which were higher than those in our results. Luan et al. [23] compared ST-ssGBLUP and MT-ssGBLUP using five traits (slaughter percentage, food consumption from 40 to 120 kg, days from 40 to 120 kg, age at 40 kg, and lean meat percentage) of Duroc boars. Their findings supported our results that MT accuracy was higher than ST accuracy. Karaman et al. [18] aimed to compare single- and multi-trait GBLUP methods based on 50 K haplotype data of 2200 animals from the Danish Holstein population. They explained the theory of the methods in detail. They also argued that MT accuracy is superior to ST accuracy. Shahi et al. [19] compared MT and ST predictions using harvest index, grain yield, grain number, spike partitioning index, fruiting efficiency in grains, and spike dry weight at anthesis+ 7 days of wheat. They mentioned that for all traits MT was super to ST accuracy except harvest index which ST prediction accuracy (0.31) was higher than MT prediction accuracy (0.30). The results of this study also support our findings on milk lactose (ML), showing that ST prediction accuracy was superior to MT prediction accuracy. Even though the study of Shahi et al. [19] was about plant material, both their results and our results show that ST prediction accuracy can be superior to MT prediction accuracy for such traits. Gaire et al. [22] aimed to compare single- and multi-trait GBLUP methods using days to heading, incidence, severity, *Fusarium* damaged kernels, and deoxynivalenol content of wheat. They mentioned that multi-trait prediction accuracy was superior to single-trait prediction accuracy for all of the traits they examined. Colombani et al. [30] estimated the accuracies of 0.52, 0.71 and 0.71 for milk yield, fat percentage, and protein percentage, respectively. Kemper et al. [31] compared some ST and MT methods using Holstein and Jersey cattle genomic data. They concluded that MT estimation accuracies were superior to ST estimation accuracies even though the differences can be ignored. The differences in accuracies reported among studies may have resulted from training population size, number of SNPs used, methods used, breeding pressure on the population, etc.

Our biases were higher than the results of Makgahlela et al. [29] for milk, protein, and fat yield. Obtained biases (coefficients of regression) and the measure of slope bias in terms of the variance of the GEBV relative to the adjusted phenotype, showed underestimation in this study because all values were greater than one. The biases did not vary so much among traits and methods which is supported by the results of Manzanilla-Pech et al. [32]. Calus and Veerkamp [33] developed methods for multi-trait genomic breeding value prediction, to enable multi-trait genomic selection, and to compare the accuracy of prediction among the different methods and with equivalent single-trait models, based on the results of applications to simulated datasets. Their biases for simulated data were greater than one which was proof for underestimation. Kemper et al. [31] declared that all biases from ST and MT estimations were greater than one for milk, protein, and fat yield for Holstein and Jersey cattle. Colombani et al. [30] found overestimated biases for milk yield, fat percentage and protein percentage. The differences in biases among researchers may have resulted from training population size, number of SNP’s used, and used methods.

Makgahlela et al. [29] found heritability values of 0.39, 0.31 and 0.36 for milk, protein, and fat yields, respectively. Our results were similar for the heritability of protein and lower for milk and fat yield. Kemper et al. [31] found the heritability values 0.56, 0.46 and 0.49 for milk yield, fat yield, and protein yield, respectively. These heritability values were higher than in our results. Aspilcueta-Borquis et al. [34] used MT-ssGBLUP for Brazilian buffalo milk traits. They estimated the heritability values as 0.25, 0.37 and 0.42 for 305-days milk yield, fat percentage, and protein percentage, respectively. Their results were similar to our results in terms of milk yield, but higher than our results for protein and fat percentage. Antanaitis et al. [35] estimated the heritability values of 0.231, 0.310, 0.342, and 0.431 for milk yield, fat percentage, protein percentage, and lactose percentage, respectively. Our results for milk yield were nearly the same, but their heritability values were higher than ours in term of fat percentage, protein percentage, and lactose percentage. Liu et al. [36] estimated 0.12, 0.30, and 0.32 heritability values for milk yield, fat percentage, and protein percentage, respectively, for Holstein cattle. The differences in heritability values among researchers may have resulted from the used methods, structure of the population, genetic variance, etc.

Kemper et al. [31] found a genetic correlation of 0.32 between milk fat and milk yield, 0.53 between fat and protein yields, and 0.81 between milk and protein yields. Their genetic correlation results were positive because they used protein and fat yields instead of the percentage which was used in our study. Our negative results can be understood using general dairy science knowledge. Aspilcueta-Borquis et al. [34] found a −0.35 genetic correlation between milk yield and fat percentage, −0.16 genetic correlation between milk yield and protein percentage, and 0.33 genetic correlation between fat percentage and protein percentage. Their genetic correlation results were stronger than those in our results. This may be the cause of buffalos’ lower milk yield and intensive milk composition. Dadpasand et al. [37] estimated −0.27 genetic correlation between milk yield and fat percentage, 0.37 genetic correlation between milk yield and protein percentage for Holstein cows using REML. The positive genetic correlation between milk yield and protein percentage is an unexpected result, contrary to our result and is hard to interpret. Antanaitis et al. [35] found genetic correlations in milk yield, milk fat (%), milk protein (%), and milk lactose (%) of Holstein cows using BLUP. They found −0.488 genetic correlations between milk fat percentage and milk yield, −0.399 between protein percentage and milk yields, 0.401 between lactose percentage and milk yields, 0.366 between fat and protein percentages, −0.095 between fat and lactose percentages, and 0.366 between fat and protein percentages. Their genetic correlation results were higher than our results but followed the same direction. Our results between fat and protein percentages in terms of genetic correlation were higher than theirs. The differences in genetic correlations among studies may have resulted from the structure of the population, milk yield, genotype, feeding practices, and used methods.

When the marker effect rank correlation was interpreted, it was understood that the rank relations were so weak that they could be ignored except between milk dry matter and milk fat percentage, which is positive and at an average level (0.603). This situation showed that markers have different effects on each trait. The improvement of milk fat percentage increases the milk dry matter because their affected markers were found in relation. However, multi-trait breeding aims are hard to reach.

Kemper et al.’s [31] declared marker effects shape was similar to our findings. In our results, only a small number of SNP’s had a significant effect on the traits, similar to studies by Liu et al. [36], who mentioned the milk composition of Holstein cattle.

## 5. Conclusions

Multi-trait genomic selection methods are generally more accurate than single-trait genomic selection methods. Even though differences between multi- and single-trait estimation accuracies are low enough to be ignored, using multi-trait methods has some additional benefits such as estimating genetic correlations and determining common effective SNPs for all traits. It can be said that the most important benefit brought by multi-trait estimation is the computation of genetic correlations within the genomic prediction. We found generally weak genetic correlations except between fat and protein percentages. Estimated heritability values were low for milk yield and milk compositions as expected. Marker rank correlations showed that the multi-trait breeding aims are hard to reach because our results showed that the relations among marker effects were low which means that each trait is determined by different markers and the number of common markers is not high enough to reveal a strong relationship.

## Figures and Tables

**Figure 1 animals-13-03070-f001:**
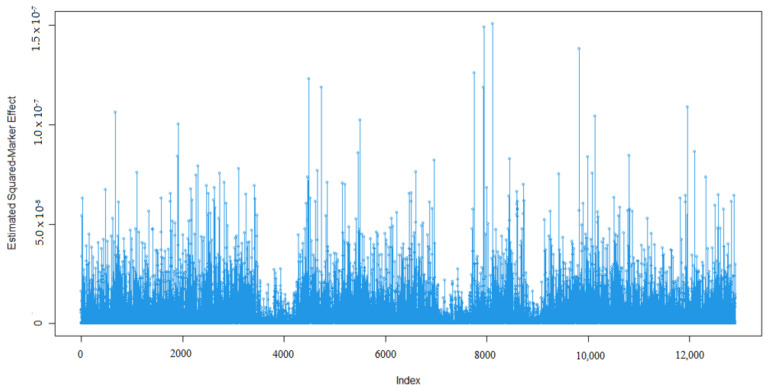
Marker effects (estimated squared marker effects) for 305-day milk yield.

**Figure 2 animals-13-03070-f002:**
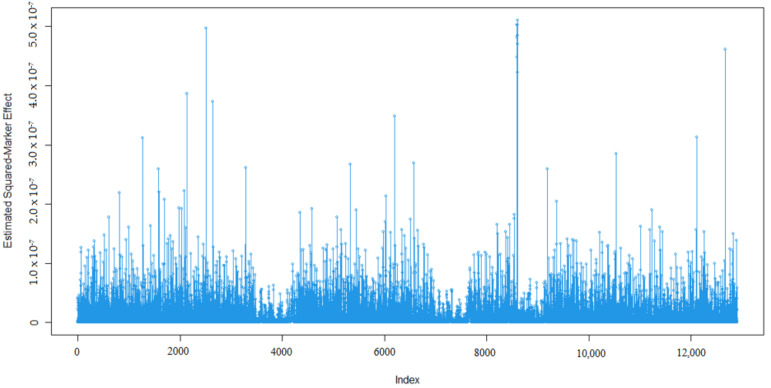
Marker effects (estimated squared marker effects) for milk fat percentage.

**Figure 3 animals-13-03070-f003:**
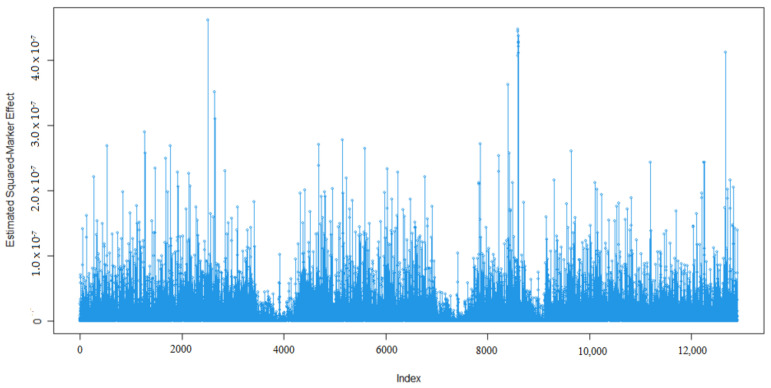
Marker effects (estimated squared marker effects) for milk protein percentage.

**Figure 4 animals-13-03070-f004:**
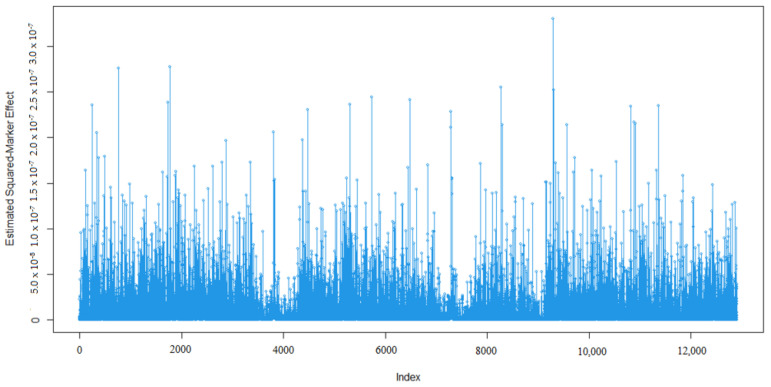
Marker effects (estimated squared marker effects) for milk lactose percentage.

**Figure 5 animals-13-03070-f005:**
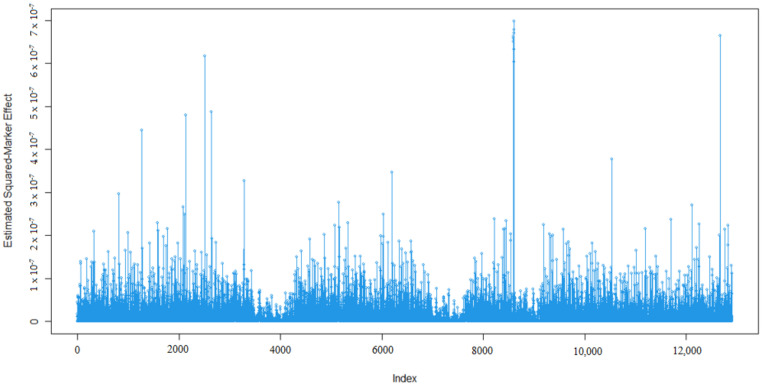
Marker effects (estimated squared marker effects) for milk dry matter percentage.

**Table 1 animals-13-03070-t001:** Spearman rank correlation coefficients of marker effects for traits of interest.

	MF	MP	ML	MDM
MY	−0.155	−0.142	0.008	−0.150
MF		0.201	0.015	0.603 **
MP			0.015	0.309 *
ML				0.072

MY = 305-days milk yield, MF = Milk fat percentage, MP = Milk protein percentage, ML = Milk lactose percentage, MDM = Milk dry matter percentage, * = *p* < 0.05, ** = *p* < 0.001.

**Table 2 animals-13-03070-t002:** Accuracy and deviations for the traits from multi-trait (MT) and single-trait (ST) predictions.

	MT		ST		Accuracy Differences (MT-ST)
	r	b	r	b	
MY	0.854	1.456	0.849	1.431	0.005
MF	0.880	1.578	0.870	1.499	0.010
MP	0.869	1.356	0.866	1.402	0.003
ML	0.770	1.102	0.773	1.127	−0.003
MDM	0.882	1.620	0.876	1.532	0.006

MY = 305-days milk yield, MF = Milk fat percentage, MP = Milk protein percentage, ML = Milk lactose percentage, MDM = Milk dry matter percentage, MT = Multi-trait estimation, ST = Single-trait estimation.

**Table 3 animals-13-03070-t003:** Genetic correlations (below diagonal), phenotypic correlations (upper diagonal), heritability and standard errors (diagonal).

	MY	MF	MP	ML	MDM
MY	0.2424 ± 0.002	−0.374 *	−0.270	−0.005	−0.362 *
MF	−0.104	0.2223 ± 0.0002	0.582 *	−0.021	0.942 **
MP	−0.092	0.132	0.3029 ± 0.003	−0.015	0.780 **
ML	0.001	0.017	0.021	0.2171 ± 0.002	0.141
MDM	−0.080	0.499 *	0.218	0.041	0.2437 ± 0.0002

MY = 305-days milk yield, MF = Milk fat percentage, MP = Milk protein percentage, ML = Milk lactose percentage, MDM = Milk dry matter percentage, * = *p* < 0.05, ** = *p* < 0.001.

## Data Availability

To access the data please contact the author H.Ö.

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
