# Peer review of "Multi-Trait Single-Step Genomic Prediction for Milk Yield and Milk Components for Polish Holstein Population"

_animals, 2023, doi:10.3390/ani13193070_

Round 1

Reviewer 1 Report

REVIEW for the journal Animals (ISSN 2076-2615)

Article “Multi-Trait Single-Step Genomic Prediction for Milk Yield and Milk Components for Polish Holstein Population

Manuscript ID: animals-2561902

Authors:  Hasan Önder, Beata Sitskowska, Burcu Kurnaz, Dariusz Piwczyński, Magdalena Kolenda, Uğur Şen, Cem Tırınk, Demet Çanga Boğa

            Brief summary.  Milk production traits play an important role in the breeding and genetic improvement of cattle. The aim of this study was to assess the predictive capability of a multi-trait genomic prediction model that considers interactions among marker effects, in order to estimate the heritability and genetic correlations of the milk production traits in the population of Polish Holstein Friesian cows. Nevertheless, the findings of this research could hold significance and relevance for animal breeders and researchers in various other countries. The comprehensive exploration of the genetic traits and correlations within the context of milk production sheds light on the intricate interplay between different factors, ultimately enriching our understanding of cattle genetics and breeding strategies.

General concept comments

1.       Introduction. In the introduction, a review of the 33 literature sources related to the analyzed topic is provided, and the objectives of the article are stated. However, I found a lack of a clear formulation of the hypothesis.

2.       Materials and Methods. The genetic material for this study was collected as part of a routine breeding value estimation. Milk recording data were gathered between 2016 and 2019 using the A4 method. The multi-trait model GBUP (Genomic Best Linear Unbiased Prediction) was employed to estimate the heritability and genetic correlations of milk production traits (milk yield, milk fat percentage, milk protein percentage, milk lactose percentage, and milk dry matter percentage) over a 305-day period. The study design is well-documented and clearly described, providing a clear outline of how the research was conducted.

3.       Data management and statistical evaluation. I omitted a detailed description of all the statistical methods utilized, which takes into consideration the hypotheses and objectives outlined in the article.

4.       I believe that the analysis of the results presented in the article aligns with the objectives and goals of the research.

5.       The conclusions drawn from this study are in complete alignment with its intended objectives. The comprehensive analysis and findings of the research substantiate the initial aims, providing a robust framework for understanding the relationships and dynamics among the milk production traits in the Polish Holstein Friesian cow population. The study successfully demonstrates how the multi-trait genomic prediction model, which takes into consideration the intricate interactions among marker effects, effectively estimates both heritability and genetic correlations among the key traits. Furthermore, the implications of these conclusions extend beyond the immediate scope of the study. The insights gained from this research have the potential to contribute significantly to the field of cattle breeding and genetics, not only in Poland but also in other countries with similar cattle populations. The methodologies and approaches employed in this study could serve as valuable references for animal breeders and scientists globally, aiding them in making informed decisions for enhanced breeding programs and genetic improvements.

Specific comments

1.       I suggest that the authors expand the section on statistical methods in the methodology of the article. Not all statistical analysis methods and indicators used in the results were described in the methodology (e.g., Spearman rank correlation coefficient, phenotypic correlations, standard error of heritability).

2.       It would be advisable to improve the clarity and comprehensibility of Figure 1.

Conclusion.  The feedback I have offered is of a minor nature, and I trust that it will contribute to enhancing the overall quality of this engaging and valuable article.

Sincerely, reviewer.

Author Response

Dear reviewer,

Thank you for your valuable comments that give us a chance to improve the manuscript.

Comment 1

Introduction: “However, I found a lack of a clear formulation of the hypothesis.”

Answer 1

The sentence “The hypothesis of this study was the multi-trait genomic prediction model more accurate than single-trait genomic prediction model.” Was added to the end of Introduction section.

Comment 2

I omitted a detailed description of all the statistical methods utilized, which takes into consideration the hypotheses and objectives outlined in the article.

Answer 2

We add subsections

2.1. Materials

2.2. Methods

2.2.1. Quality control and data preparation

2.2.2. Data analysis

2.2.3. Single-trait and multi-trait genomic prediction models

and we add the text “To examine the relationship among marker effects impact level of traits, Spearman rank correlation analysis was used. To examine the relationship among phenotypic correlations of the traits Pearson correlation analysis were used. Genetic correlation between trait t1 and t2 was calculated as  where  is the genetic variance–covariance matrix for multiple traits. The  was calculated as , where var(SNPi) is the genotype variance for SNPi and  is the estimated marker effect vector for SNPi in iteration k for an analysis run over k2 iterations and with k1 burn-in iterations [29]. ]. Standard error of the heritability was calculated from the iterations [20].”

Comment 3

I suggest that the authors expand the section on statistical methods in the methodology of the article. Not all statistical analysis methods and indicators used in the results were described in the methodology (e.g., Spearman rank correlation coefficient, phenotypic correlations, standard error of heritability).

Answer 3

Given in Answer 2

Comment 4

It would be advisable to improve the clarity and comprehensibility of Figure 1.

Answer 4

Figure 1 is separated to Figure 1, 2, 3, 4, and 5 to be more readable.

Reviewer 2 Report

Dear Editor and Authors,

I send you my review about the article entitled “Multi-Trait Single-Step Genomic Prediction for Milk Yield and Milk Components for Polish Holstein Population”.

The aim of the paper, as reported in the scope was to evaluate the predictive ability of a multi-trait genomic prediction model that accounts for interactions between marker effects, to estimate heritability and genetic correlations, of some production traits in the Polish Holstein Friesian cows population.

In my opinion, although the Article is well written, in a good English language and it is well structured, it show, also, some lacks that I reported below.

The introduction it too much long and dispersive, thus it should be summarise.

Moreover, always in the introduction should be better explained the originality of this Article. To improve this aspect, I suggest to the Authors, to citing the articles that have investigated similar aspects of the ones of their research and, after, stress the difference among these research and their article.

The paragraph materials and methods result complete, however, in this paragraph should be reported some detail of samples collected as for examples form how many herd they were collected or in what seasons.

Furthermore, should be the reported methods used for the determination of milk paramethers.

Moreover, to the data analysis should dedicated a paragraph.

The results is very well presented and they are well discussed, also in comparison to the data reported in the literature.

Moreover, also, data shown in tables and figures result complete.

On the other hand the conclusions could be to be too synthetic respect the data showed and to the aim of the research, in my opinion, this paragraph it should best stress the relevance of the findings of this research.

Best regards

Author Response

Dear reviewer,

Thank you for your valuable comments that give us a chance to improve the manuscript.

Comment 1

The introduction it too much long and dispersive, thus it should be summarise.

Answer 1

Some sentenced in the Introduction section were deleted.

Comment 2

Moreover, always in the introduction should be better explained the originality of this Article. To improve this aspect, I suggest to the Authors, to citing the articles that have investigated similar aspects of the ones of their research and, after, stress the difference among these research and their article.

Answer 2

It was mentioned in the original draft as

“A great majority of the genomic selection studies [25-28] has focused on single-trait breeding value estimation. However, multi-trait methods are expected to yield more accurate predictions than single-trait methods as in traditional BLUP. Simulations have shown that multi-trait genomic prediction can lead to a considerable increase in genomic prediction accuracy [29-32].”

End of the above sentences we add “Jia and Jannink [29] compared multi-trait (MT) and single-trait (ST) estimation accuracies and mentioned that MT and ST methods showed similar prediction accuracies when Guo et al. [30] reported that the multi-trait predictions can improve the accuracies of single-trait predictions for genomic prediction.”

Comment 3

The paragraph materials and methods result complete, however, in this paragraph should be reported some detail of samples collected as for examples form how many herd they were collected or in what seasons.

Answer 3

 Some corrections were done

“reared in private enterprises” à “reared in 12 different private enterprises”.

“and farms (12) were used” à“and farms (1,2,…,12) were used”

Unfortunately our data has no complete season information.

Comment 4

Furthermore, should be the reported methods used for the determination of milk paramethers.

Moreover, to the data analysis should dedicated a paragraph.

Answer 4

The sentence was changes as “The data regarding milk performance was collected during routine test days by the Polish Federation of Cattle Breeders and Dairy Farmers (PFCB&DF). Authors were granted access to the data. The description provided in the manuscript is in accordance with PFCB&DF description.” Because we did not collect data using any automatic tool. We took the data from the Polish Federation of Cattle Breeders and Dairy Farmers

Comment 5

On the other hand the conclusions could be to be too synthetic respect the data showed and to the aim of the research, in my opinion, this paragraph it should best stress the relevance of the findings of this research.

Answer 5

The conclusion section was extended.

Reviewer 3 Report

Rewiew Manuscript ID: animals-2561902

Brief summary: The paper entitled “Multi-Trait Single-Step Genomic Prediction for Milk Yield and Milk Components for Polish Holstein Population” investigates the predictive ability of a multi-trait genomic prediction model to estimate the heritability and genetic correlations of the traits 305 days milk yield, milk fat %, milk protein %, milk lactose % and milk dry matter %, in the Polish Holstein population during a period of four-year observation period (2016-2019). A total of 14,742 SNP genotype records from 586 Polish Holstein Friesian dairy cattle were used. A single-trait-ssGBLUP (ST) and a multi-trait-ssGBLUP (MT) estimation method were used for the study. The authors found the highest heritability value for protein (0.3029) and the lowest heritability value for lactose (0.2171). The estimated heritability values were low for milk yield and milk composition, while the highest genetic correlation was estimated between dry matter and fat (0.4990). In conclusion, multi-trait genomic prediction may be more beneficial than single-trait genomic prediction.

The paper is in line with the topic of the journal, but in its current form it needs minor revision.

Introduction: It is very broad and include many bibliographical references.

Materials and methods: this section needs to be improved, in particular it is necessary to describe the tools used to determine the milk quality parameters should be described in detail. Unfortunately the somatic cell count (health parameter) was not include which I suggest to evaluated for the future study.

The results and the discussion are well structured but need some improvements, especially in the figures.

Conclusions need to be expanded.

References are complete with d.o.i. 

Specific comments (the line number is missing in the text, I will refer to the page and text):

Page 1/12 (Abstract): Keywords: Multi-trait prediction; Single-trait prediction; Milk composition; GBLUP; Genomic selection; Holstein; Poland.

Check “Holstein; Poland”.

Page 2/12 (Introduction): The nutritional value of cattle milk is rich in essential nutrients such as protein, cal-cium, phosphorus, and vitamins B12 and D. These nutrients are important for healthy bone growth, muscle development, and overall health [8,9].

Check “Regarding "cow's milk protein" and "human health diseases". I suggest the authors briefly mention the adverse effects related to "cow's milk protein allergy" and the effects related to "beta-casein variants" (A1, A2, etc).”

Page 3/12: Genomic selection is a breeding technique used in the dairy cattle industry to improve the genetic potential of cattle for milk production and other desirable traits.

Check “Is genomic selection a breeding technique used only in the dairy cattle industry??”

Page 4/12 (Material and Methods): 305-day milk yield (MY, kg), milk fat percentage (MF, %), milk protein percentage (MP, %), milk lactose percentage (ML, %), milk dry matter percentage (MDM, %), and 14,742 SNP genotype records (after selection on same genes) for 586 Polish Holstein Friesian (PHF) dairy cattle reared in private enterprises in Poland were used.

Check “305-day”, rewrite as “Three hundred and five days”, or similar form.

Page 4/12: Milk recording data were collected in the period from 2016 to 2019 according to A4 method (the interval between two successive recordings of two daily milking ranging be-tween 28 and 33 days) accredited by the International Committee for Animal Recording (ICAR), by the Milk Analysis Laboratory of the Polish Federation of Cattle Breeders and Dairy Farmers (PFCB&DF) which is certified by ISO 17025 and certified by ICAR.

Check “The standard "ISO/IEC 17025", which codifies the general requirements for the competence of test and calibration laboratories, is too general and insufficient in the description of this paragraph. I suggest that the authors include the automated tool used to determine milk quality parameters (fat, protein, lactose, TS) in addition to the quality control system."

Page 5/12: 400 animals were used as a training and the remaining 97 as a test population. The lactation order (OL, 1, 2 and 3) and farms (12) were used as a fixed factor and days in milk (DIM) as a random factor. The single trait GBUP can be defined as given in Equation 1; 

Check “400 animals”, rewrite as “Four hundred” or similar form.

Check “GBUP”

Page 5/12: The random effect design matrices are equal for each trait. a is a vector of pre-dicted genetic values of the individuals for all traits with a~N(0, ⊗K), and e is the error vector with e~N(0, R⊗?), where K is the realized additive relationship matrix among in-dividuals estimated from the markers,  and R are the unstructured variance–covariance matrices for the genetic and residual effects between traits, respectively [35,36].

Check “Further details, such as correlations (Spearman) and accuracy, should be included at the end of this paragraph.”

Page 5-6/12 (Results): Figure 1. Marker effects (estimated squared marker effects): a) 305-day milk yield, b) Milk fat percentage, c) Milk protein percentage, d) Milk lactose percentage, e) Milk dry matter percentage. 

Check “The 5 figures need to be enlarged and improved, including the x-axis legends and the y-axis,  which is currently illegible”.

Page 6/12: 3.2. Accuracy. The prediction accuracy was defined as the correlation between the true breeding values (observed phenotype data) and the predicted GEBV values in the validation pop-ulation [20]. The accuracy (r) and deviations (b) for the traits from multi-trait (MT) and single-trait (ST) predictions were calculated and are presented in Table 2. 

Check “See the penultimate comment before this one”.

Page 8/12 (Discussion): Shahi et al. [32] compared MT and ST predictions using harvest index, grain yield, grain number, spike partitioning index, fruit-ing efficiency in grains, and spike dry weight at anthesis+ 7 days of wheat. They men-tioned that for all traits MT was super to ST accuracy except harvest index which ST pre-diction accuracy (0.31) was higher than MT prediction accuracy (0.30). The results of this study also support our findings on milk lactose (ML), showing that ST prediction accuracy was superior to MT prediction accuracy. 

Check “The reference to "Shahi et al. [32]", should be predictions on plant material and not animal material, should be better contextualised”.

Page 9/12 (Conclusion): Check “The conclusions are too brief, I suggest the authors to expand them”.

The paper requires moderate editing of the English language.

Author Response

Dear reviewer,

Thank you for your valuable comments that give us a chance to improve the manuscript.

Comment 1

Page 1/12 (Abstract)Keywords: Multi-trait prediction; Single-trait prediction; Milk composition; GBLUP; Genomic selection; Holstein; Poland.

Answer 1

 The keywords “Holstein” and “Poland” were deleted. You are right these are so general terms.

Comment 2

 Page 2/12 (Introduction): The nutritional value of cattle milk is rich in essential nutrients such as protein, cal-cium, phosphorus, and vitamins B12 and D. These nutrients are important for healthy bone growth, muscle development, and overall health [8,9].

Check “Regarding "cow's milk protein" and "human health diseases". I suggest the authors briefly mention the adverse effects related to "cow's milk protein allergy" and the effects related to "beta-casein variants" (A1, A2, etc).”

Answer 2

According to the other referee’ comments this paragraph was deleted because the comments were the Introduction section is too long.

Comment 3

Page 3/12: Genomic selection is a breeding technique used in the dairy cattle industry to improve the genetic potential of cattle for milk production and other desirable traits.

Check “Is genomic selection a breeding technique used only in the dairy cattle industry??”

Answer 3

Corrected as “Genomic selection is a breeding technique used in the dairy cattle industry (as in other species for the same or different aims) to improve the genetic potential of cattle for milk production and other desirable traits.”

Comment 4

Page 4/12 (Material and Methods): 305-day milk yield (MY, kg), milk fat percentage (MF, %), milk protein percentage (MP, %), milk lactose percentage (ML, %), milk dry matter percentage (MDM, %), and 14,742 SNP genotype records (after selection on same genes) for 586 Polish Holstein Friesian (PHF) dairy cattle reared in private enterprises in Poland were used.

Check “305-day”, rewrite as “Three hundred and five days”, or similar form.

Answer 4

All “305-day” expressions were changed as “305-days” to be in similar form.

Comment 5

Page 4/12: Milk recording data were collected in the period from 2016 to 2019 according to A4 method (the interval between two successive recordings of two daily milking ranging be-tween 28 and 33 days) accredited by the International Committee for Animal Recording (ICAR), by the Milk Analysis Laboratory of the Polish Federation of Cattle Breeders and Dairy Farmers (PFCB&DF) which is certified by ISO 17025 and certified by ICAR.

Check “The standard "ISO/IEC 17025", which codifies the general requirements for the competence of test and calibration laboratories, is too general and insufficient in the description of this paragraph. I suggest that the authors include the automated tool used to determine milk quality parameters (fat, protein, lactose, TS) in addition to the quality control system."

Answer 5

The sentence was changes as “The data regarding milk performance was collected during routine test days by the Polish Federation of Cattle Breeders and Dairy Farmers (PFCB&DF). Authors were granted access to the data. The description provided in the manuscript is in accordance with PFCB&DF description.” Because we did not collect data using any automatic tool. We took the data from the Polish Federation of Cattle Breeders and Dairy Farmers

Comment 6

Page 5/12: 400 animals were used as a training and the remaining 97 as a test population. The lactation order (OL, 1, 2 and 3) and farms (12) were used as a fixed factor and days in milk (DIM) as a random factor. The single trait GBUP can be defined as given in Equation 1;

Check “400 animals”, rewrite as “Four hundred” or similar form.

Check “GBUP”

Answer 6

Corrected as “400 animals were used as a training population set and the remaining 97 animals as a test population set.”

“GBUP” was corrected as “GBLUP”

Comment 7

Page 5/12: The random effect design matrices are equal for each trait. a is a vector of pre-dicted genetic values of the individuals for all traits with a~N(0, ⊗K), and e is the error vector with e~N(0, R⊗?), where K is the realized additive relationship matrix among in-dividuals estimated from the markers,  and R are the unstructured variance–covariance matrices for the genetic and residual effects between traits, respectively [35,36].

Check “Further details, such as correlations (Spearman) and accuracy, should be included at the end of this paragraph.”

Answer 7

The sentence “The prediction accuracy was defined as the Pearson correlation between the true breeding values (observed phenotype data) and the predicted GEBV values in the validation population [20].” Was removed from the Results section to the Method Section.

Comment 8

Page 5-6/12 (Results): Figure 1. Marker effects (estimated squared marker effects): a) 305-day milk yield, b) Milk fat percentage, c) Milk protein percentage, d) Milk lactose percentage, e) Milk dry matter percentage.

Check “The 5 figures need to be enlarged and improved, including the x-axis legends and the y-axis,  which is currently illegible”.

Answer 8

Figure 1 is separated to Figure 1, 2, 3, 4, and 5 to be more readable.

Comment 9

Page 6/12: 3.2. Accuracy. The prediction accuracy was defined as the correlation between the true breeding values (observed phenotype data) and the predicted GEBV values in the validation pop-ulation [20]. The accuracy (r) and deviations (b) for the traits from multi-trait (MT) and single-trait (ST) predictions were calculated and are presented in Table 2.

Check “See the penultimate comment before this one”.

Answer 9

Please see Answer 7.

Comment 10

Page 8/12 (Discussion): Shahi et al. [32] compared MT and ST predictions using harvest index, grain yield, grain number, spike partitioning index, fruit-ing efficiency in grains, and spike dry weight at anthesis+ 7 days of wheat. They men-tioned that for all traits MT was super to ST accuracy except harvest index which ST pre-diction accuracy (0.31) was higher than MT prediction accuracy (0.30). The results of this study also support our findings on milk lactose (ML), showing that ST prediction accuracy was superior to MT prediction accuracy.

Check “The reference to "Shahi et al. [32]", should be predictions on plant material and not animal material, should be better contextualised”.

Answer 10

This sentences was extended as “Shahi et al. [32] compared MT and ST predictions using harvest index, grain yield, grain number, spike partitioning index, fruiting efficiency in grains, and spike dry weight at anthesis+ 7 days of wheat. They mentioned that for all traits MT was super to ST accuracy except harvest index which ST prediction accuracy (0.31) was higher than MT prediction accuracy (0.30). The results of this study also support our findings on milk lactose (ML), showing that ST prediction accuracy was superior to MT prediction accuracy even the study of Shahi et al. [32] was about plant material, both their and our results showed that ST prediction accuracy can be superior to MT prediction accuracy for such traits.”

Comment 11

Page 9/12 (Conclusion): Check “The conclusions are too brief, I suggest the authors to expand them”.

Answer 11

The conclusion section was extended.

Reviewer 4 Report

In order to improve the quality of the manuscript, I ask the authors to consider the following comments:

-        Unfortunately, I received the manuscript without line numbering, so I cannot name a specific line.

-        The introduction contains some information that has been repeated several times. I recommend that authors avoid such repetitive information. For example, the ingredients of cow's milk, the importance of milk for nutrition and the cultural value of cow's milk.

-        In the materials and methods, it was mentioned that 400 animals were used as training animals and the remaining 97 were used as test population. The question is: What does 400 animals mean as training animals?

-         The authors did not mention whether the cows used came from one or more farms.

-        Normally, a separate section under Materials and methods was written for statistical analysis and this section is missing here in this manuscript.

-        Unfortunately, the results in Figure 1 have very small fonts and numbers, making them unreadable.

-        In Table 1 and 3, the correlation coefficient between the examined parameters was presented without specifying the significance. I therefore recommend that the authors use the significant value in addition to the correlation value for all parameters that have significance.

-        Regarding the reference list, there are some references that do not have a complete page number. That means they have a beginning and no end. These are the following numbers: 2,3,6,21,28,30,32,33,40,44,46,48 and 49. In addition, all references' data were in bold, only two references (4 and 11) were not in bold.

Author Response

Dear reviewer,

Thank you for your valuable comments that give us a chance to improve the manuscript.

Comment 1

The introduction contains some information that has been repeated several times. I recommend that authors avoid such repetitive information. For example, the ingredients of cow's milk, the importance of milk for nutrition and the cultural value of cow's milk.

Answer 1

 The Introduction section was shortened.

Comment 2

In the materials and methods, it was mentioned that 400 animals were used as training animals and the remaining 97 were used as test population. The question is: What does 400 animals mean as training animals?

Answer 2

A sentence “For the genomic prediction the population divided into two groups as train and test population. The records of the training set was used to build up the estimation model and the records of the test set (not included in the estimation) was used to predict their GEBVto estimate accuracies [20].” Was added to Data analysis section.

Comment 3

The authors did not mention whether the cows used came from one or more farms.

Answer 3

 Some corrections were done

“reared in private enterprises” à “reared in 12 different private enterprises”.

“and farms (12) were used” à“and farms (1,2,…,12) were used”

Comment 4

 Normally, a separate section under Materials and methods was written for statistical analysis and this section is missing here in this manuscript.

Answer 4

We add subsections

2.1. Materials

2.2. Methods

2.2.1. Quality control and data preparation

2.2.2. Data analysis

2.2.3. Single-trait and multi-trait genomic prediction models

Comment 5

Unfortunately, the results in Figure 1 have very small fonts and numbers, making them unreadable.

Answer 5

Figure 1 is separated to Figure 1, 2, 3, 4, and 5 to be more readable.

Comment 6

In Table 1 and 3, the correlation coefficient between the examined parameters was presented without specifying the significance. I therefore recommend that the authors use the significant value in addition to the correlation value for all parameters that have significance.

Answer 6

Corrected.

Comment 7

Regarding the reference list, there are some references that do not have a complete page number. That means they have a beginning and no end. These are the following numbers: 2,3,6,21,28,30,32,33,40,44,46,48 and 49. In addition, all references' data were in bold, only two references (4 and 11) were not in bold.

Answer 7

The date of references 4 and 11 was bolded.

Some references hasn’t page number because they used article ID in their system.
